# Cognitive Orientation to Daily Occupational Performance: A Randomized Controlled Trial Examining Intervention Effects on Children with Developmental Coordination Disorder Traits

**DOI:** 10.3390/brainsci13050721

**Published:** 2023-04-25

**Authors:** Masanori Yasunaga, Hideki Miyaguchi, Chinami Ishizuki, Yosuke Kita, Akio Nakai

**Affiliations:** 1Health and Counseling Center, Campus Life Health Support/Consultation Center, Osaka University, 3rd Floor, Student Exchange Building, 1-10 Machikaneyama-cho, Toyonaka 560-0043, Japan; 2Department of Human Behavior Science of Occupational Therapy, Health Sciences Major, Graduate School of Biomedical & Health Sciences, Hiroshima University, 1-2-3 Minamiku Kasumi, Hiroshima 734-8551, Japan; hmiya@hiroshima-u.ac.jp (H.M.);; 3Department of Psychology, Faculty of Letters, Keio University, 2-15-45 Mita, Minato-ku, Tokyo 108-8345, Japan; 4Cognitive Brain Research Unit (CBRU), Faculty of Medicine, University of Helsinki, 3B Haartmaninkatu, 00014 Helsinki, Finland; 5Graduate School of Clinical Education & The Center for the Study of Child Development, Institute for Education, Mukogawa Women’s University, 6-46 Ikebiraki, Nishinomiya 663-8558, Japan

**Keywords:** children, developmental coordination disorder, cognitive orientation to daily occupational performance, motor skills, performance skills

## Abstract

Children with traits of developmental coordination disorder (DCD-t) may experience occupational performance problems that go unrecognized and therefore may not be adequately supported. The cognitive orientation to daily occupational performance (CO-OP) approach has been effective in interventions for developmental coordination disorder (DCD). Based on an open-label, randomized controlled trial design, this study evaluated the effects of CO-OP on the occupational performance and motor skills of older kindergarten children with DCD-t using the School Assessment of Motor and Process Skills (S-AMPS) and the Movement Assessment Battery for Children, Second Edition. Children with a total DCDQ score of less than 40 or M-ABC2 scores in the 5th to 16th percentile were considered to have DCD-t. Furthermore, children with DCD-t and S-AMPS process skills less than 0.7 were considered to have DAMP (Deficits in Attention, Motor control and Perception)-t. After 3 months of CO-OP intervention, the performance and motor skills of children with DCD-t improved significantly. However, there were no significant changes noted in the motor skills of children with DAMP-t, although their occupational performance improved. These results suggest that CO-OP is effective even for older kindergarten children with DCD-t. However, further improvement of the CO-OP approach or a different strategy is required for children with ADHD comorbidity.

## 1. Introduction

Developmental coordination disorder (DCD) is usually diagnosed through standardized tests that assess the acquisition and execution of coordinated motor skills based on the Diagnostic and Statistical Manual of Mental Disorders (DSM-5) [1]. A diagnosis is made when a child’s test results are considerably below the expected level for a particular age, given general opportunities for skill learning. Problems related to these motor skills will eventually lead to a child experiencing difficulties with the movements required for various aspects of activities of daily living (ADLs), thereby reducing the frequency of their participation in such activities [2]. Delayed learning of motor skills and movements for ADLs has been shown to be related to low executive functioning [3]. Moreover, when there is a delay in learning motor skills, children reportedly suffer from reduced self-efficacy [4] and may also be unable to attend school. Children with low motor performance often suffer from bullying [5,6] and mental health disorders [7]. Thus, there is a growing need for early detection [8] and intervention [9] for children with DCD.

Studies on the relationship between the motor skills of children with DCD and their social and cognitive capabilities suggest that problems related to executive function (EF) impede the acquisition of motor skills [10,11,12]. EF is classified into three categories: suppression, shifting, and updating [13]; therefore, these assessments must also be considered. Tests on working memory (WM) in children with DCD and attention-deficit hyperactivity disorder (ADHD) have revealed significantly lower WM scores among these children compared to healthy children [14,15]. Children with ADHD are also reported to have difficulty planning their actions and executing or performing various tasks [16].

Therefore, considering the characteristics of the deficits in attention, motor control, and perception collectively defined as the DAMP syndrome [17], and referring to a combination of DCD and ADHD, it is necessary to devise assessments and interventions that focus on the problem of motor impairments and EF.

The international clinical practice recommendations (ICPR-DCD) [18] advocate using the Movement Assessment Battery for Children (Second Edition) (M-ABC2) [19] as a tool for assessing DCD, including the DCD Questionnaire (DCDQ) [20,21] as a screening tool. The M-ABC2’s total scores in the 5th percentile and 15th percentile are set as the cut-off values. Children with ≤5th percentile scores have severe motor dysfunction, and those with scores ranging from the 6th to the 15th percentiles are in the risk group [19].

Tests that assess occupational performance include the Assessment of Motor and Process Skills (AMPS) [22] and the School Assessment of Motor and Process Skills (S-AMPS) [23]. The AMPS sub-items are mainly divided into motor and process skills. Process skills are reported to be closely related to attention and parts of memory [24], and are correlated with WM [25]. A previous study showed that conducting the S-AMPS to children with ADHD generated significantly lower scores than those for neurotypical children for all subscales of process skills [26].

Therefore, the AMPS and S-AMPS are not tests of EF, but rather assess the target child’s occupational performance and participation while they are engaged in activities. They contain elements of assessment that check EF from the perspective of the target child’s (or adult’s) scenarios of everyday life and learning. Therefore, we consider the S-AMPS results to be appropriate as an important outcome measure when assessing older kindergarten children who are at risk of developing DCD and ADHD.

Cognitive orientation to daily occupational performance (CO-OP) [27] is one of the interventions related to the activity-oriented and participation-oriented approaches for DCD, as recommended by the International Clinical Practice [18]. The CO-OP approach is based on cognitive behavioral therapy. The CO-OP consists of seven elements [27]: the goal chosen by the client, dynamic performance analysis, use of cognitive strategies, guided discoveries, principles of enablement, parental participation, and form of intervention. Two important aspects of this intervention must be noted [27]. The first is the global strategy (GS), which introduces the child to the framework of “goal, plan, do, and check” for problem-solving. The second is the domain-specific strategy (DSS), which helps the child to acquire specific skills for solving each problem (classified into seven categories and characterized by the use of verbal guidance). CO-OP is an approach developed for DCD, but it has also been applied to people with autism and cerebral palsy [28] to improve their occupational performance. Evidence on its use for DAMP characteristics is limited, and findings on its effects on motor skills have been mixed [29]. Therefore, the purpose of this study is to evaluate occupational performance and motor skills of older kindergarten children with developmental coordination disorder traits (DCD-t) and ADHD co-morbidity and to verify the effect of a CO-OP intervention on children with these conditions.

The significance of this study lies is its attempt to demonstrate the effectiveness of CO-OP intervention in improving occupational performance and motor skills not only in children with DCD, but also in older kindergarten children with DCD-t, a condition present in approximately 10% of DCD cases [30]. This suggests that more children with coordination problems may benefit from the CO-OP approach. In addition, a difference noted in the effect of CO-OP between children with DCD-t and DAMP traits (DAMP-t) may indicate the need for intervention according to the co-morbidity of other neurodevelopmental disorders such as ADHD.

## 2. Materials and Methods

### 2.1. Study Design

This study employed an open-label, randomized controlled trial design. Previous studies have emphasized the importance of efforts in early detection [8] and intervention [9] to support children with DCD. According to the *DSM-5* and ICPR-DCD, a diagnosis of DCD should be made after the age of five years [4]; thus, older kindergarten children aged 5–6 years were included in this study.

### 2.2. Participants and Procedures

Of an initial sample comprising 181 children (from six classes of five private kindergartens), 103 children with parental approval were screened. The DCDQ-Japanese version (DCDQ-J) enforces the exclusion of those with severe intellectual or physical disabilities who have considerable difficulty communicating with others.

Based on this screening process, 28 children with total DCDQ-J scores of <40 points met our study’s inclusion criteria. The 28 children were randomly assigned to two groups to prevent bias. Random numbers were generated using Excel; odd numbers were assigned to the control group (n = 14) and even numbers to the intervention group (n = 14). The threshold of a total score of ≤40 points in the DCDQ-J was based on the data from five-year-old boys [22].

### 2.3. Assessments

The S-AMPS is the only standardized evaluation method designed to assess the quality of student performance in school tasks [31]. It evaluates the quality of performance without focusing on social, physical, cognitive, or psychological functions or diagnoses. Unlike other assessments, this is a four-stage assessment of students’ motor skills during occupational performance (physical effort during occupational performance) and process skills (efficiency during occupational performance). The target age was three years or older, including older kindergarten students involved in school tasks. The cut-off values were set to logits of 2.0 and 1.0 for motor skills and process skills, respectively, which were indicators of whether the participants adapted to the task in the class. Additionally, the reliability and validity of the S-AMPS were confirmed [31,32].

The M-ABC2 [19] assesses the delay in children’s motor skills, and the eight subtests are classified into three subcategories: manual dexterity (three items), aiming and catching (two items), and balance (three items). In addition to the total standardized score, the test provides a standardized score for each subcategory. The test takes approximately 20–40 min to complete. Total scores in the 5th percentile and 15th percentile were set as cut-off values. Children with ≤5th percentile scores had severe motor dysfunction, and those with scores ranging from the 6th to the 15th percentiles were in the at-risk group [8]. The M-ABC2 was employed based on recommendations made in international guidelines [19], and its reliability and validity have been confirmed [33]. Since standardization of the Japanese version of the M-ABC2 [34,35] is still in progress, the score was calculated using original data from the United Kingdom.

The DCDQ is a screening test developed and standardized according to international guidelines presented by Nakai et al. [36] in collaboration with Wilson et al. [20], who created the original version of the test. The test is appropriate for use among children ranging from 5 to 15 years of age. It consists of 15 items and three subscales: control during movement (six items), fine motor/handwriting (four items), and general coordination (five items). The participants were asked to respond to each item on a five-point Likert scale based on comparisons with other children of the same age. The maximum score was 75 points. The lower the score, the more severe the related DCD issue. A total score of ≤5% meets the diagnostic level of DCD, and a score of 6–15% indicates the possibility of DCD. The reliability and validity of the DCDQ-J, as well as its high sensitivity and specificity, have been confirmed [21,37].

### 2.4. Intervention

The duration of the intervention was eight weeks, from June to July 2017 and from January to February 2018. Interventions using the CO-OP approach with older children with DCD-t were performed once a week for 40 min, for a total of eight sessions. The rationale for setting the session duration to 40 min per intervention was based on previous research using the CO-OP approach with children aged 5–7 years.

Two occupational therapists (with 10 and 15 years of experience) attended the CO-OP workshop in Japan and conducted the intervention. The activities selected by the participants in the intervention group were mainly of 10 types. Among these activities, the most commonly selected were handicraft activities using craft scissors, writing exercises, and playing catch.

#### 2.4.1. First Session

During the 0–35-min time bracket after the children carried out their respective activities, they were individually coached on particular strategies to use, such as the cognitive strategy from the CO-OP theory. The strategy chosen was based on the results of the dynamic performance analysis and S-AMPS. During the 35–40-min time bracket following this step, the areas where the children were able to succeed in the activities and where they could devise strategies were confirmed. They were also instructed on the activities to be conducted during their kindergarten breaks.

#### 2.4.2. Second to Eighth Sessions

During the 0–5-min time bracket, we reviewed the plan devised by each child during the previous session. During the 5–35 min time bracket, the activities selected by each child were performed using cognitive strategies (GS and DSS). During the ensuing 35–40-min time bracket, the experiment was conducted in the same manner as in the first session. In all the interventions, the intended activities of each child were recorded using a video camera for approximately 10 min, and the child and therapist reviewed the recording. Children in the control group were not subjected to any intervention and spent their time at the kindergarten.

The assessments used to determine the intervention effect in the two groups were the S-AMPS and M-ABC2. The assessments were performed before the intervention (baseline), after the intervention (post), and three months after the end of the intervention (post three months; Figure 1).

### 2.5. Ethical Considerations

Ethical approval was obtained from the Hiroshima University Clinical Research Ethics Committee (approval no. C-173). The study was verbally explained to the principal teachers of the participating kindergartens, and the teachers’ written consent was obtained.

Subsequently, the teachers distributed written documents to the parents and obtained their consent. Occupational therapists used illustrations and other materials to explain the study to the participating children and obtained their consent orally.

### 2.6. Statistical Analysis

For the pre-assessments, the Mann–Whitney U test was used to compare the baseline measurements between the intervention and control groups. In addition, normality was assessed using the Shapiro–Wilk test. As the results of the S-AMPS did not show a normal distribution, the Friedman test was performed for the baseline, post, and post-three-month assessments in the two groups. Subsequently, the Wilcoxon signed-rank test (Bonferroni method) was performed for multiple comparisons in the intervention group. *p* < 0.016 was considered significant.

The children from our study sample were classified into the DAMP-t group (process skills scores < 0.7) and DCD-t group. The Mann–Whitney U test was used to compare the degree of change in the three assessments (baseline, post, and post-three-month assessments) between the intervention and control groups. All analyses were conducted using IBM SPSS Statistics (Version 26).

## 3. Results

Of the 28 DCD-t children, 17 had below-average total scores for five-year-olds in the S-AMPS process skills dimension [38] and the M-ABC2 [19], 11 had lower than −1 SD scores in S-AMPS process skills [38], and five had lower than −1 SD scores in the M-ABC2 [19].

A comparison of the results of the S-AMPS with the measured values in 5–12-year-old children [31] revealed that their motor skill scores were close to those characterizing typical development, and their process skill scores were close to those found in mild neurodevelopmental disorders. The baseline values did not show significant differences between the two groups in terms of age, sex, and DCDQ, S-AMPS, and M-ABC2 scores (Table 1).

The correlation coefficients of the DCDQ-J, S-AMPS, and M-ABC2 were calculated (Table 2). The S-AMPS motor and process skills were positively correlated with the DCDQ fine motor/handwriting skills.

A significant difference was observed in the S-AMPS process skill scores between the baseline and post assessments in the intervention group (Table 3).

Using Cohen’s d for multiple comparisons [39], the effect size of the intervention group for the baseline and post assessments was 0.78, and that of the intervention group for the baseline and post-three-month assessments was 0.60, showing a large effect. The change in the S-AMPS motor skills scores was not significantly different among the three assessments (Table 3). However, a significant difference was noted in the M-ABC2 total score in the baseline and post-three-month assessments of the intervention group (Table 3).

The effect size of the total score was large (0.75) for the baseline and post-three-month assessments, and moderate (0.52) for the baseline and post assessments. Similar to Heus et al.’s [40] study, the present study found that the baseline and post-three-month assessments showed a larger effect than the post-intervention evaluation of children with DCD (effect size: 0.42).

In the control group, no significant difference was observed between the baseline assessment and the baseline and post-three-month assessments (Table 3).

The study participants were classified into the DAMP-t group (process skills scores < 0.7) and DCD-t group, and the amount of change in the three assessments was compared between the two groups (Table 4). The results revealed a significant difference only in the baseline and post assessments. The DAMP-t group showed a significant difference only with the intervention group in S-AMPS process skills. The DCD-t group showed a significant difference only in the intervention group in relation to S-AMPS process skills, DCDQ-J fine motor/handwriting abilities, and M-ABC2 manual dexterity skills.

## 4. Discussion

In this study, we tested the hypothesis that interventions using CO-OP in older kindergarten children with DCD-t would be effective in improving their motor skills by enhancing their occupational performance through task-specific EFs. At baseline, from the scores of older kindergarten children with DCD-t, we observed a correlation between S-AMPS process skills and DCDQ-J fine motor/handwriting skills, with approximately 60% showing a delay in occupational performance (DAMP-t). The results of the intervention revealed greater changes in EF between the baseline and post-intervention in the intervention group compared to the control group, and improvements in motor skills were noted between the baseline and post-three-month values. A difference in the intervention effects was also observed, which was related to the presence or absence of DAMP-t in children with DCD-t. The findings are discussed in detail below.

### 4.1. Relationship between Occupational Performance and Motor Skills

Our findings revealed that part of the EF of older kindergarten children with DCD-t was related to their motor skills, especially finger dexterity. Prior research has reported correlations between EF and WM in children with DCD-t aged 3–5 years [3], and that preschoolers with DCD have problems with WM and attention inhibition control [41]. Moreover, during their elementary school years, 40–60% of DCD children show impaired EF [12,42]. Children for whom DCD persisted showed lower EF compared to neurotypical and remission-type DCD children (cases in which DCD symptoms had been alleviated for two years after being diagnosed with DCD) [40]. Past studies that investigated these relationships between DCD and ADHD showed poor EF in the target populations [20,43,44]. These studies appeared to have assessed problems with EF by targeting children with DCD and ADHD; however, they did not assess ADHD traits. Therefore, we believe that it is necessary to use the S-AMPS before primary school to assess DCD and examine elements of executive functions, together with motor skills, to gain a comprehensive understanding of the skill status of each child assessed.

### 4.2. Advantages of Considering S-AMPS Results as an Outcome Measure for Older Kindergarten Children with DCD-t or DAMP-t

Reportedly, 30–50% [45,46] of children with ADHD have co-morbid DCD. Furthermore, lower motor coordination has been found to be associated with increased inattention and hyperactivity [47]. Leonard et al. [11] reported that DCD children with concurrent traits of ADHD suffer not only from a delay in motor skills development but also experience problems with executive function that affect nonverbal areas of visual WM and fluency of cognition. Additionally, AMPS process skills are related to attention and certain aspects of memory [23], and they are correlated with WM [25]. Therefore, it appears that S-AMPS process skills contain elements that make it possible to observe “Working Memory capabilities (hereinafter ‘WM’), cognitive flexibility, and inhibitory control,” which are part of EFs ([48], p. 298). According to the International Clinical Practice Recommendations, interventions for children with DCD must focus on their activity and participation levels and must aim to increase their participation in physical activities [9,18]. The S-AMPS makes it possible to observe instances of occupational performance during kindergarten and school activities by considering the respective environmental contexts, children, and tasks [31], and it can also conduct assessments that take the level of activity participation into account.

Accordingly, we suggest that the S-AMPS is an important assessment tool for evaluating activity and participation levels in children prone to DCD-t and DAMP-t, as it enables the assessment of EF elements from an occupational performance perspective.

### 4.3. Intervention Effects of CO-OP in Older Kindergarten Children with DCD-t

The following discussion details our observations of children’s attention levels at the early stages of learning and how their occupational performance was enhanced by using the Guide Discovery strategy (e.g., asking questions instead of giving directions) based on the CO-OP theory (GS) and a cognitive strategy (DSS).

In the early learning stages, children with DCD show impairments in the dorsolateral prefrontal cortex [49] and corpus callosum [50], which are areas of the brain related to attention. This is likely the reason for their problems with attention control and interventions [51]. In the handwriting tasks, the children in the intervention group left tools and materials scattered on their desks and were unable to recognize spatial arrangements. They were also unable to discover the DSS using the Guide Discovery strategy, likely due to poor WM functioning and an insufficient understanding of the process needed to reach goals and the steps needed to execute movements. Therefore, the occupational therapists assisted them in the activities by presenting options to the children and making them aware of the options.

The children subsequently understood that tools and materials should be placed in the corner of the desk as much as possible to make it easier to turn their attention to the activity using the DSS, enabling them to tackle the tasks more smoothly. Björkdahl et al. [25] reported that AMPS process skills are significantly correlated with WM. The fact that process skills improved between baseline and post-intervention in the current study suggests that a change had occurred in the EFs.

In their intervention study using CO-OP, Araujo et al. [52] reported a significant difference in EF (cognitive flexibility and inhibitory control), indicating that enhancing performance skills (part of EF) showed results similar to those of previous studies. Based on these prior findings, in the current study, the CO-OP theory’s DSS was employed among children in the intervention group who were coached in aspects related to awareness, such as paying attention to their environment during a task to achieve their goals. This appeared to have led to the enhancement of process skills related to WM, which is part of EFs.

### 4.4. Difference in Intervention Effects When Children with DCD-t Were Divided into the DAMP-t and DCD-t Groups

Based on the results of the intervention conducted among 28 children with DCD-t, we found that those in the DAMP-t group displayed only enhanced occupational performance, while those in the DCD-t group showed improvements in both occupational performance and motor skills (Table 4). We present the following two reasons for the lack of significant differences in the motor skills of children in the DAMP-t group:Although performance skills have been enhanced, it is difficult to generalize these skills to other motor skills and to rapidly acquire new motor skills.Many sub-items in the M-ABC2 assess the speed of physical activities but not the quality of performance.

Regarding point (1) presented above, Izadi-Najafabadi et al. [51] reported that children with DCD and ADHD have a low-functioning frontoparietal network, which impedes the automation of movements in some cases. Similarly, children with DAMP-t may require more time to reach the automation stage of motor skill acquisition [53,54]. Compared with cases in which scores for motor skills increased as a result of CO-OP intervention [55,56], this study had a lower intervention frequency and was of a shorter duration. Moreover, no homework was assigned; therefore, we were unable to increase the opportunities for the children to study the strategy (Guide Discovery) outside the 40-minute intervention session. Compared to children with only DCD, those with DAMP-t showed no radical changes that might result in improvements in their motor skills via changes in relevant brain regions [51]. Children with DAMP-t required more time to recognize and unify motor skill learning than children with DCD-t. Children with DAMP-t may have failed to reach the automation stage; consequently, the performance skills that they had acquired may not have led to any generalization of motor skills.

Izadi-Najafabadi et al. [57] reported that children with DCD showed increased functional connectivity between the default mode network and the right anterior cingulate gyrus after CO-OP intervention. Furthermore, their three-month follow-up after the intervention showed additional improvements, with increased functional connectivity between the dorsal attention network and precentral gyrus. For children with DCD, increased functional connectivity in networks related to the self, emotion, and attention regulation underlie the improvements in motor skills observed after CO-OP interventions.

In contrast, children with comorbid DCD and ADHD (the “DAMP syndrome”) showed no changes in brain functional connectivity after CO-OP intervention. Modifications to the CO-OP protocol are required to induce brain functional connectivity in children with both DCD and ADHD.

With respect to point (2) presented above, children with DAMP-t need to consciously focus their attention while engaging in activities during the early stages of learning, unlike children with DCD [57], which might affect their performance speed.

Manual dexterity, a subscale of the M-ABC2 that shows significant differences in children with DCD, emphasizes the speed of physical exercise over accuracy. Therefore, it is likely that even if their unique execution skill capabilities for a particular task had improved, children with DAMP-t lacked performance speed and consequently failed to raise their M-ABC2 scores. Thus, we infer that children with DAMP-t did not show significant differences in their motor skills after the intervention. The reason for this may be that task-specific execution skills did not generalize to other motor skills and were not reflected in the results of the M-ABC2 test, as the children spent more time on the motor skills they had not practiced.

Children with DCD with ADHD characteristics experience EF problems, such as excessive or sudden loss of attention [58] and problems with attention inhibition [59]. In the future, it would be valuable to investigate whether DCD-t children have EF impairments before conducting any interventions.

Despite some of its notable findings, this study has some limitations that must be acknowledged. First, the researchers involved in the interventions conducted in this study were also involved in the assessments. In future interventional research, the evaluators should be blinded. Second, the main measures of CO-OP [60] and the Performance Quality Rating Scale (PQRS) [27,61] were not used. As the performance skills of children with DCD-t changed after the intervention in this study, the PQRS may be used as an auxiliary tool to evaluate the CO-OP intervention outcomes of future studies. Third, performance and motor skills constituted the only DCD-t ascertained in the 28 child participants. Fourth, other important individuals (e.g., homeroom teachers or parents) did not participate in this study. In future research, it would be valuable to conduct assessments that focus on the ADHD rating scale, EF [62], and WM [63], including the participation of teachers and parents. This should be done to incorporate opportunities for children to learn various skills using Guide Discovery and DSS, and to investigate whether the resulting acquired skills can be generalized.

## 5. Conclusions

Based on an open-label, randomized controlled design, this study revealed that older kindergarten children with DCD-t experienced difficulties with occupational performance and motor skills. In addition, three months of CO-OP intervention improved occupational performance and motor skills in children with DCD-t. Meanwhile, children with DAMP-t showed improvements in occupational performance but not exercise performance. These results clarified the role that CO-OP plays in improving occupational performance and motor skills not only in children with DCD, but also in those with DCD-t, which is more common among older kindergarten-aged children. However, the effects of CO-OP were different in children with co-morbid ADHD co-morbidity, suggesting that a suitable intervention strategy based on the situation is necessary.

## Figures and Tables

**Figure 1 brainsci-13-00721-f001:**
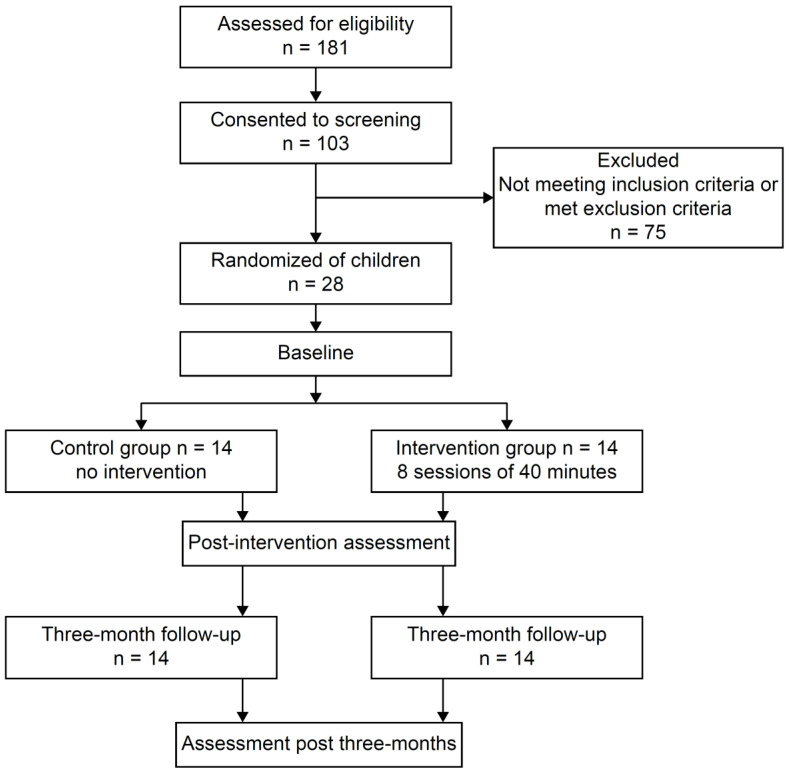
Study flow diagram based on the Consolidated Standards of Reporting Trials (CONSORT).

**Table 1 brainsci-13-00721-t001:** Baseline comparison between the intervention and control groups.

		Baseline Assessment			
Outcome measure	Intervention group		Control group		
	(n = 14)		(n = 14)		
	Mean (SD)	Median	Mean	Median	*p*-value
		(25th–75th percentile)	(SD)	(25th–75th percentile)	
Sex	9 boys, 5 girls		9 boys, 5 girls		0.13 ^†^
Age	70.00 (5.23)	70 (64.0–75.25)	68.93 (3.91)	69.50 (65.75–71.25)	0.53 ^‡^
DCDQ-J					
Total score	37.90 (0.98)	39 (34.40–40.0)	37.93 (0.518)	38.5 (36–40)	0.95 ^‡^
CDM	13.57 (0.98)	13.5 (10.75–17.0)	14.29 (0.90)	14 (11.75–17.00)	0.60 ^§^
FM/HW	10.14 (0.61)	10.5 (8.0–12.0)	10.57 (0.69)	10 (8–12.5)	0.80 ^‡^
GC	13.86 (0.44)	14 (12.75–15.0)	13.07 (0.53)	13 (11.75–14.25)	0.27 ^§^
School AMPS motor skill	2.5 (0.11)	2.50 (2.25–2.83)	2.34 (0.11)	2.35 (1.91–2.63)	0.34 ^§^
School AMPS process skill	0.54 (0.10)	0.48 (0.30–0.75)	0.54 (0.14)	0.50 (0.08–1.10)	0.98 ^§^
M-ABC2					
Total score	74.14 (2.50)	74.00 (64.75–81.25)	75.50 (1.71)	76.0 (69.75–79.50)	0.66 ^§^
MD	29.79 (1.08)	31.0 (25.75–33.25)	29.07 (0.92)	29.50 (35.75–31.0)	0.62 ^§^
AC	13.79 (1.27)	12.0 (10.0–17.75)	14.0 (0.86)	13.5 (11.0–17.0)	0.57 ^‡^
Bal	32.9 (1.40)	31.5 (28.75–35.25)	32.29 (0.87)	32.50 (29.0–35.0)	0.55 ^§^

DCDQ, Developmental Coordination Disorder Questionnaire; AMPS, Assessment of Motor and Process Skills; M-ABC2, Movement Assessment Battery for Children–Second Edition; CDM, control during movement; FM/HW, fine motor/handwriting; GC, general coordination; MD, manual dexterity; AC, aiming and catching; Bal, balance; ^†^ chi-square test for goodness of fit; ^‡^ Mann–Whitney U test; ^§^ two-sample *t*-test.

**Table 2 brainsci-13-00721-t002:** Relationship between the S-AMPS, Developmental Coordination Disorder Questionnaire, and Movement Assessment Battery for Children (Second Edition).

	D-Total	D-CDM	D-FM	D-GC	S-Motor	S-Process	M-Total	M-MD	M-AC	M-FM
D-Total	―									
D-CDM	0.63 *	―								
D-FM	0.31	−0.38	―							
D-GC	0.20	−0.13	0.00	―						
S-Motor	0.29	−0.19	0.60 *	−0.03	―					
S-Process	0.19	−0.35	0.63 *	0.22	0.65 *	―				
M-Total	−0.08	−0.09	0.14	0.00	−0.40	−0.11	―			
M-MD	−0.04	−0.24	0.25	0.12	−0.20	−0.06	0.72 *	―		
M-AC	−0.04	0.11	−0.12	−0.08	−0.44	−0.30	0.68 *	0.31	―	
M-Bal	−0.21	−0.13	0.05	−0.06	−0.26	0.11	0.72 *	0.26	0.27	―

* *p* < 0.01. D, Developmental Coordination Disorder Questionnaire; S, School Assessment of Motor and Process Skills; M, Movement Assessment Battery; CDM, control during movement; FM, fine motor/handwriting; GC, general coordination; MD, manual dexterity; AC, aiming and catching; Bal, balance.

**Table 3 brainsci-13-00721-t003:** Median and standard deviation values of the School Assessment of Motor and Process Skills and Movement Assessment Battery for Children–Second Edition at baseline, post, and post-three-months testing.

			Baseline		Post				Post Three Months			
			Median (25th–75th percentiles)	SD	Median (25th–75th percentiles)	SD	P ^(a)^	Effect size	Median (25th–75th percentiles)	SD	P ^(b)^	Effect size
Assessment		n										
School AMPS motor skill	Intervention	14	2.50 (2.25–2.83)	0.43	2.50 (2.23–2.83)	0.34	0.75	0.08	2.68 (2.31–2.80)	0.39	0.48	0.19
	Control	14	2.35 (1.91–2.63)	0.41	2.25 (1.89–2.51)	0.33	0.43	0.21	2.6 (2.28–2.80)	0.41	0.21	0.34
School AMPS process skill	Intervention	14	0.48 (0.30–0.73)	0.38	0.89 (0.80–1.15)	0.35	0.004 *	0.78	0.79 (0.53–1.03)	0.39	0.03	0.60
	Control	14	0.50 (0.08–1.10)	0.53	0.5 (0.25–0.68)	0.33	0.51	0.18	0.65 (0.25–0.80)	0.38	0.03	0.13
M-ABC2 total score	Intervention	14	74.00 (64.8–81.25)	9.36	81.0 (74.0–86.5)	7.73	0.05	0.52	82.0 (76.5–92.5)	8.08	0.005 *	0.75
	Control	14	76.0 (69.75–79.50)	6.39	74.0 (65.50–81.25)	9.70	0.59	0.14	79.0 (70.50–89.25	12.79	0.47	0.20
M-ABC2 MD	Intervention	14	31.0 (25.75–33.25)	4.04	29.0 (26.50–34.0)	5.34	0.92	0.03	31.50 (28.75–34.25)	4.28	0.46	0.20
	Control	14	29.50 (35.75–31.0)	3.40	26.0 (20.75–34.25)	7.03	0.43	0.21	32.0 (29.25–36.0)	6.32	0.03	0.57
M-ABC2 AC	Intervention	14	12.0 (10.0–17.75)	4.74	16.0 (13.75–20.50)	3.83	0.13	0.40	16.0 (14.75–23.25)	4.78	0.02	0.65
	Control	14	13.5 (11.0–17.0)	3.23	14.0 (13.0–16.0)	3.39	0.84	0.50	15.5 (11.0–19.50)	4.27	0.45	0.20
M-ABC2 Bal	Intervention	14	31.5 (28.75–35.25)	5.25	33.0 (30.0–36.25)	2.85	0.16	0.38	34.50 (33.0–36.0)	2.41	0.03	0.57
	Control	14	32.50 (29.0–35.0)	3.25	31.5 (30.75–35.25)	3.43	0.92	0.92	32.0 (29.25–36.0)	4.22	0.65	0.65

^(a)^ baseline: post; ^(b)^ baseline: post three months; * mean difference (Wilcoxon) was considered significant at 0.016. MD, manual dexterity; AC, aiming and catching; Bal, balance.

**Table 4 brainsci-13-00721-t004:** Changes in scores on the School-Assessment of Motor and Process Skills, Movement Assessment Battery for Children–Second Edition, and DCDQ-J at baseline and post assessment.

	DAMP-t			DCD-t		
	Intervention Group	Control Group	*p*-Value	Intervention Group	Control Group	*p*-Value
	(n = 8)	(n = 9)		(n = 6)	(n = 5)	
	Mean ± SD	Mean ± SD		Mean ± SD	Mean ± SD	
DCDQ-J						
Total score	8.5 ± 10.13	5.56 ± 7.88	0.511 ^§^	9 ± 4.10	0.8 ± 6.10	0.52 ^‡^
CDM	5 ± 5.40	1.78 ± 4.47	0.321 ^‡^	4.83 ± 3.37	2.4 ± 5.03	0.363 ^§^
FM/HW	2.13 ± 4.61	2.22 ± 3.77	0.962 ^§^	2.17 ± 1.94	−2.4 ± 2.70	0.01 ^§^*
GC	0.63 ± 3.42	1.56 ± 1.01	0.48 ^§^	1.67 ± 2.16	0.8 ± 3.96	0.655 ^§^
School AMPS	0.35 ± 0.35	0.078 ± 0.47	0.198 ^§^	−0.32 ± 0.43	−0.48 ± 0.40	0.542 ^§^
motor skill						
School AMPS	0.54 ± 0.31	0.119 ± 0.40	0.021 ^‡^*	0.298 ± 0.45	−0.46 ± 0.24	0.008 ^§^**
process skill						
M-ABC2						
Total score	4.13 ± 10.73	−1 ± 11.49	0.359 ^§^	8.67 ± 9.03	−0.24 ± 8.88	0.72 ^§^
MD	−1.5 ± 5.56	−2 ± 7.35	0.879 ^§^	2.5 ± 3.89	−1.4 ± 9.61	0.038 ^§^*
AC	1.25 ± 6.78	1.0 ± 4.82	0.931 ^§^	5.33 ± 6.77	−0.6 ± 4.83	0.126 ^‡^
Bal	3.13 ± 6.66	0 ± 2.35	0.241 ^§^	0.83 ± 2.99	0 ± 3.67	0.688 ^§^

DCD-t, developmental coordination disorder trait; DAMP-t, Deficits in attention, motor control, and perception trait; DCDQ-J, Developmental Coordination Disorder Questionnaire Japanese version; School AMPS, School Assessment of Motor and Process Skills; MABC-2, Movement Assessment Battery for Children-2; CDM, control during movement; FM/HW, fine motor/handwriting; GC, general coordination; MD, manual dexterity; AC, aiming and catching; Bal, balance; ^‡^ Mann–Whitney U test; ^§^ two-sample *t*-test; * *p* < 0.005, ** *p* < 0.001.

## Data Availability

The study data can be accessed via the following: https://doi.org/10.6084/m9.figshare.22246978 assessed on 13 March 2023.

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
