# Peer review of "Cognitive Orientation to Daily Occupational Performance: A Randomized Controlled Trial Examining Intervention Effects on Children with Developmental Coordination Disorder Traits"

_brainsci, 2023, doi:10.3390/brainsci13050721_

Round 1

Reviewer 1 Report

Thank you for the opportunity to review this manuscript entitled "Effects of Cognitive Orientation to Daily Occupational Performance in Children with Developmental Coordination Disorder Traits"

There are several comments for the authors:

- The title does not summarize well the type of study and its objective, reformulate

- Introduction: better organize the information and make it make a more logical sense about the population, the evaluation and finally the objective. There are Paragraphs that are summaries, not introductions (eg lines 90-101). In the introduction, the importance of the study should not be highlighted, that is up for discussion, rather it is necessary to justify why it is done and the objective of it, an aspect that is missing.

- Include Institutional Review Board Statement also in material and methods.

Where is the sample collected from? Specify.

- Figure 1 and tables: N is always in lower case.

- The conclusions must be consistent with the objectives.

Author Response

Dear reviewer

We become indebted to. My name is Masanori Yasunaga.

I will answer your questions with your advice.
Thank you

Masanori Yasunaga

Reviewer 2 Report

Manuscript Title: Effects of Cognitive Orientation to Daily Occupational Performance in Children with Developmental Coordination Disorder Traits.

Overview: this study tested the hypothesis that CO-OP (occupational performance) has an intervention effect in performance and motor skills of older kindergarten children with DCD (developmental coordination disorder) traits through an unblinded randomized controlled trial.  

General comments: This is an interesting manuscript that addresses an important area of unmet medical need. Please see my specific comments below for more details.

This open-label randomized controlled trial, consists of a non-structured abstract with 6 keywords, 5 sections (introduction, methods with 6 subsections (and one of them, subsection 4 with 2 subsections), results, discussion and conclusions) on 14 pages of single-spaced text with embedded figures. There are 65 references, 4 tables and 1 figure.

Approved by the ethics board of the Hiroshima University Clinical Research Ethics Committee (No.C-173-27/4-2017). The study was conducted in accordance with the Declaration of Helsinki. And informed consent was obtained from all study participants.

Specific comments:

  1. The keywords are absolutely fine.
  2. Introduction: I think this section can be a bit summarized, it needs to be less extensive (lines 37 to 135). It is a very important issue and must be specified with the most relevant information
  3. Methods: the methodology is complete, widely described.
  4. Results: the results are clearly expressed, with 4 tables. I would like the authors to review table 1, too much data and difficult to follow.
  5. Discussion: is correct, adapting to results obtained.

Author Response

(The authors gave the same response as above.)

Round 2

Reviewer 2 Report

Thank you for the changes, you have done a good job.